# Introspective Classification with Convolutional Nets

**Long Jin**
UC San Diego
longjin@ucsd.edu

**Justin Lazarow**
UC San Diego
jlazarow@ucsd.edu

**Zhuowen Tu**
UC San Diego
ztu@ucsd.edu

## Abstract

We propose introspective convolutional networks (ICN) that emphasize the importance of having convolutional neural networks empowered with generative capabilities. We employ a reclassification-by-synthesis algorithm to perform training using a formulation stemmed from the Bayes theory. Our ICN tries to iteratively: (1) synthesize pseudo-negative samples; and (2) enhance itself by improving the classification. The single CNN classifier learned is at the same time generative — being able to directly synthesize new samples within its own discriminative model. We conduct experiments on benchmark datasets including MNIST, CIFAR-10, and SVHN using state-of-the-art CNN architectures, and observe improved classification results.

## 1 Introduction

Great success has been achieved in obtaining powerful discriminative classifiers via supervised training, such as decision trees [34], support vector machines [42], neural networks [23], boosting [7], and random forests [2]. However, recent studies reveal that even modern classifiers like deep convolutional neural networks [20] still make mistakes that look absurd to humans [11]. A common way to improve the classification performance is by using more data, in particular "hard examples", to train the classifier. Different types of approaches have been proposed in the past including bootstrapping [31], active learning [37], semi-supervised learning [51], and data augmentation [20]. However, the approaches above utilize data samples that are either already present in the given training set, or additionally created by humans or separate algorithms.

In this paper, we focus on improving convolutional neural networks by endowing them with synthesis capabilities to make them internally generative. In the past, attempts have been made to build connections between generative models and discriminative classifiers [8, 27, 41, 15]. In [44], a self supervised boosting algorithm was proposed to train a boosting algorithm by sequentially learning weak classifiers using the given data and self-generated negative samples; the generative via discriminative learning work in [40] generalizes the concept that unsupervised generative modeling can be accomplished by learning a sequence of discriminative classifiers via self-generated pseudo-negatives. Inspired by [44, 40] in which self-generated samples are utilized, as well as recent success in deep learning [20, 9], we propose here an introspective convolutional network (ICN) classifier and study how its internal generative aspect can benefit CNN's discriminative classification task. There is a recent line of work using a discriminator to help with an external generator, generative adversarial networks (GAN) [10], which is different from our objective here. We aim at building a *single CNN model* that is simultaneously discriminative and generative.

The introspective convolutional networks (ICN) being introduced here have a number of properties. (1) We introduce introspection to convolutional neural networks and show its significance in supervised classification. (2) A reclassification-by-synthesis algorithm is devised to train ICN by iteratively augmenting the negative samples and updating the classifier. (3) A stochastic gradient descent sampling process is adopted to perform efficient synthesis for ICN. (4) We propose a supervised formulation to directly train a multi-class ICN classifier. We show consistent improvement over state-of-the-art CNN classifiers (ResNet [12]) on benchmark datasets in the experiments.

## 2 Related work

Our ICN method is directly related to the generative via discriminative learning framework [40]. It also has connection to the self-supervised learning method [44], which is focused on density estimation by combining weak classifiers. Previous algorithms connecting generative modeling with discriminative classification [8, 27, 41, 15] fall in the category of hybrid models that are direct combinations of the two. Some existing works on introspective learning [22, 3, 38] have a different scope to the problem being tackled here. Other generative modeling schemes such as MiniMax entropy [50], inducing features [6], auto-encoder [1], and recent CNN-based generative modeling approaches [48, 47] are not for discriminative classification and they do not have a single model that is both generative and discriminative. Below we discuss the two methods most related to ICN, namely generative via discriminative learning (GDL) [40] and generative adversarial networks (GAN) [10].

**Relationship with generative via discriminative learning (GDL) [40]**

ICN is largely inspired by GDL and it follows a similar pipeline developed in [40]. However, there is also a large improvement of ICN to GDL, which is summarized below.

- **CNN vs. Boosting**. ICN builds on top of convolutional neural networks (CNN) by explicitly revealing the introspectiveness of CNN whereas GDL adopts the boosting algorithm [7].

- **Supervised classification vs. unsupervised modeling**. ICN focuses on the supervised classification task with competitive results on benchmark datasets whereas GDL was originally applied to generative modeling and its power for the classification task itself was not addressed.

- **SGD sampling vs. Gibbs sampling**. ICN carries efficient SGD sampling for synthesis through backpropagation which is much more efficient than the Gibbs sampling strategy used in GDL.

- **Single CNN vs. Cascade of classifiers**. ICN maintains a single CNN classifier whereas GDL consists of a sequence of boosting classifiers.

- **Automatic feature learning vs. manually specified features**. ICN has greater representational power due to the end-to-end training of CNN whereas GDL relies on manually designed features.

**Comparison with Generative Adversarial Networks (GANs) [10]**

Recent efforts in adversarial learning [10] are also very interesting and worth comparing with.

- **Introspective vs. adversarial**. ICN emphasizes being introspective by synthesizing samples from its own classifier while GAN focuses on adversarial — using a distinct discriminator to guide the generator.

- **Supervised classification vs. unsupervised modeling**. The main focus of ICN is to develop a classifier with introspection to improve the supervised classification task whereas GAN is mostly for building high-quality generative models under unsupervised learning.

- **Single model vs. two separate models**. ICN retains a CNN discriminator that is itself a generator whereas GAN maintains two models, a generator and a discriminator, with the discriminator in GAN trained to classify between "real" (given) and "fake" (generated by the generator) samples.

- **Reclassification-by-synthesis vs. minimax**. ICN engages an iterative procedure, reclassification-by-synthesis, stemmed from the Bayes theory whereas GAN has a minimax objective function to optimize. Training an ICN classifier is the same as that for the standard CNN.

- **Multi-class formulation**. In a GAN-family work [36], a semi-supervised learning task is devised by adding an additional "not-real" class to the standard k classes in multi-class classification; this results in a different setting to the standard multi-class classification with additional model parameters. ICN instead, aims directly at the supervised multi-class classification task by maintaining the same parameter setting within the softmax function without additional model parameters.

Later developments alongside GAN [35, 36, 49, 3] share some similar aspects to GAN, which also do not achieve the same goal as ICN does. Since the discriminator in GAN is not meant to perform the generic two-class/multi-class classification task, some special settings for semi-supervised learning [10, 35, 49, 3, 36] were created. ICN instead has a single model that is both generative and discriminative, and thus, an improvement to ICN's generator leads to a direct means to ameliorate its discriminator. Other work like [11] was motivated from an observation that adding small perturbations to an image leads to classification errors that are absurd to humans; their approach is however taken by augmenting positive samples from existing input whereas ICN is able to synthesize new samples from scratch. A recent work proposed in [21] is in the same family of ICN, but [21] focuses on unsupervised image modeling using a cascade of CNNs.

# 3 Method

The pipeline of ICN is shown in Figure 1, which has an immediate improvement over GDL [40] in several aspects that have been described in the previous section. One particular gain of ICN is its representation power and efficient sampling process through backpropagation as a variational sampling strategy.

## 3.1 Formulation

We start the discussion by introducing the basic formulation and borrow the notation from [40]. Let $\mathbf{x}$ be a data sample (vector) and $y \in \{-1, +1\}$ be its label, indicating either a negative or a positive sample (in multi-class classification $y \in \{1, ..., K\}$). We study binary classification first. A discriminative classifier computes $p(y|\mathbf{x})$, the probability of $\mathbf{x}$ being positive or negative. $p(y = -1|\mathbf{x}) + p(y = +1|\mathbf{x}) = 1$. A generative model instead models $p(y, \mathbf{x}) = p(\mathbf{x}|y)p(y)$, which captures the underlying generation process of $\mathbf{x}$ for class $y$. In binary classification, positive samples are of primary interest. Under the Bayes rule:

$$p(\mathbf{x}|y = +1) = \frac{p(y = +1|\mathbf{x})p(y = -1)}{p(y = -1|\mathbf{x})p(y = +1)}p(\mathbf{x}|y = -1), \tag{1}$$

which can be further simplified when assuming equal priors $p(y = +1) = p(y = -1)$:

$$p(\mathbf{x}|y = +1) = \frac{p(y = +1|\mathbf{x})}{1 - p(y = +1|\mathbf{x})}p(\mathbf{x}|y = -1). \tag{2}$$

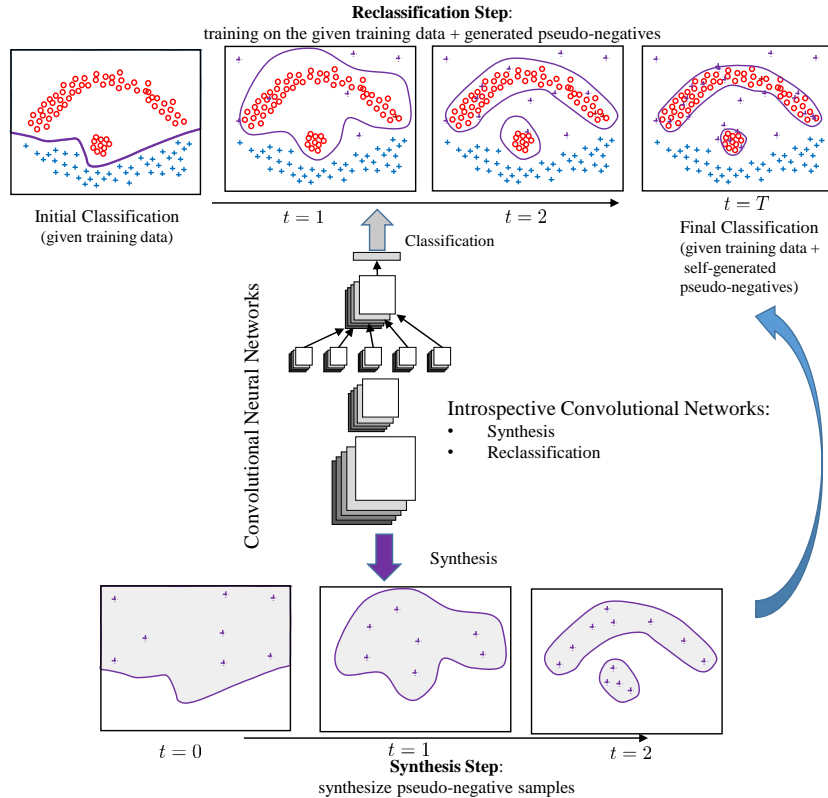

Figure 1: Schematic illustration of our reclassification-by-synthesis algorithm for ICN training. The top-left figure shows the input training samples where the circles in red are positive samples and the crosses in blue are the negatives. The bottom figures are the samples progressively self-generated by the classifier in the synthesis steps and the top figures show the decision boundaries (in purple) progressively updated in the reclassification steps. Pseudo-negatives (purple crosses) are gradually generated and help tighten the decision boundaries.

We make two interesting and important observations from Eqn. (2): 1) $p(\mathbf{x}|y = +1)$ is dependent on the faithfulness of $p(\mathbf{x}|y = -1)$, and 2) a classifier $C$ to report $p(y = +1|\mathbf{x})$ can be made **simultaneously generative and discriminative**. However, there is a requirement: having an informative distribution for the negatives $p(\mathbf{x}|y = -1)$ such that samples drawn $\mathbf{x} \sim p(\mathbf{x}|y = -1)$

have good coverage to the entire space of $\mathbf{x} \in \mathbb{R}^m$, especially for samples that are close to the positives $\mathbf{x} \sim p(\mathbf{x}|y = +1)$, to allow the classifier to faithfully learn $p(y = +1|\mathbf{x})$. There seems to exist a dilemma. In supervised learning, we are only given a set of limited amount of training data, and a classifier $C$ is only focused on the decision boundary to separate the given samples and the classification on the unseen data may not be accurate. This can be seen from the top left plot in Figure 1. This motivates us to implement the synthesis part within learning — make a learned discriminative classifier generate samples that pass its own classification and see how different these generated samples are to the given positive samples. This allows us to attain a single model that has two aspects at the same time: a generative model for the positive samples and an improved classifier for the classification.

Suppose we are given a training set $S = \{(\mathbf{x}_i, y_i), i = 1..n\}$ and $\mathbf{x} \in \mathbb{R}^m$ and $y \in \{-1, +1\}$. One can directly train a discriminative classifier $C$, e.g. a convolutional neural networks [23] to learn $p(y = +1|\mathbf{x})$, which is always an approximation due to various reasons including insufficient training samples, generalization error, and classifier limitations. Previous attempts to improve classification by data augmentation were mostly done to add more positive samples [20, 11]; we instead argue the importance of *adding more negative samples* to improve the classification performance. The dilemma is that $S = \{(\mathbf{x}_i, y_i), i = 1..n\}$ is limited to the given data. For clarity, we now use $p^-(\mathbf{x})$ to represent $p(\mathbf{x}|y = -1)$. Our goal is to augment the negative training set by generating confusing pseudo-negatives to improve the classification (note that in the end pseudo-negative samples drawn $\mathbf{x} \sim p_t^-(\mathbf{x})$ will become hard to distinguish from the given positive samples. Cross-validation can be used to determine when using more pseudo-negatives is not reducing the validation error). We call the samples drawn from $\mathbf{x} \sim p_t^-(\mathbf{x})$ **pseudo-negatives** (defined in [40]). We expand $S = \{(\mathbf{x}_i, y_i), i = 1..n\}$ by $S_e^t = S \cup S_{pn}^t$, where $S_{pn}^0 = \emptyset$ and for $t \geq 1$
$$S_{pn}^t = \{(\mathbf{x}_i, -1), i = n + 1, ..., n + tl\}.$$

$S_{pn}^t$ includes all the pseudo-negative samples self-generated from our model up to time $t$. $l$ indicates the number of pseudo-negatives generated at each round. We define a reference distribution $p_r^-(\mathbf{x}) = U(\mathbf{x})$, where $U(\mathbf{x})$ is a Gaussian distribution (e.g. $\mathcal{N}(0.0, 0.3^2)$ independently). We carry out learning with $t = 0...T$ to iteratively obtain $q_t(y = +1|\mathbf{x})$ and $q_t(y = -1|\mathbf{x})$ by updating classifier $C^t$ on $S_e^t = S \cup S_{pn}^t$. The initial classifier $C^0$ on $S_e^0 = S$ reports discriminative probability $q_0(y = +1|\mathbf{x})$. The reason for using $q$ is because it is an approximation to the true $p$ due to limited samples drawn in $\mathbb{R}^m$. At each time $t$, we then compute
$$p_t^-(\mathbf{x}) = \frac{1}{Z_t} \frac{q_t(y = +1|\mathbf{x})}{q_t(y = -1|\mathbf{x})} p_r^-(\mathbf{x}), \tag{3}$$

where $Z_t = \int \frac{q_t(y=+1|\mathbf{x})}{q_t(y=-1|\mathbf{x})} p_r^-(\mathbf{x}) d\mathbf{x}$. Draw new samples $\mathbf{x}_i \sim p_t^-(\mathbf{x})$ to expand the pseudo-negative set:
$$S_{pn}^{t+1} = S_{pn}^t \cup \{(\mathbf{x}_i, -1), i = n + tl + 1, ..., n + (t + 1)l\}. \tag{4}$$

We name the specific training algorithm for our introspective convolutional network (ICN) classifier **reclassification-by-synthesis**, which is described in Algorithm 1. We adopt convolutional neural networks (CNN) classifier to build an end-to-end learning framework with an efficient sampling process (to be discussed in the next section).

## 3.2 Reclassification-by-synthesis

We present our reclassification-by-synthesis algorithm for ICN in this section. A schematic illustration is shown in Figure 1. A single CNN classifier is being trained progressively which is simultaneously a discriminator and a generator. With the pseudo-negatives being gradually generated, the classification boundary gets tightened, and hence yields an improvement to the classifier's performance. The reclassification-by-synthesis method is described in Algorithm 1. The key to the algorithm includes two steps: (1) reclassification-step, and (2) synthesis-step, which will be discussed in detail below.

### 3.2.1 Reclassification-step

The reclassification-step can be viewed as training a normal classifier on the training set $S_e^t = S \cup S_{pn}^t$ where $S = \{(\mathbf{x}_i, y_i), i = 1..n\}$ and $S_{pn}^0 = \emptyset$. $S_{pn}^t = \{(\mathbf{x}_i, -1), i = n + 1, ..., n + tl\}$ for $t \geq 1$. We use CNN as our base classifier. When training a classifier $C^t$ on $S_e^t$, we denote the parameters to be learned in $C^t$ by a high-dimensional vector $\mathsf{W}_t = (\mathbf{w}_t^{(0)}, \mathbf{w}_t^{(1)})$ which might consist of millions of parameters. $\mathbf{w}_t^{(1)}$ denotes the weights of the top layer combining the features $\phi(\mathbf{x}; \mathbf{w}_t^{(0)})$ and $\mathbf{w}_t^{(0)}$

carries all the internal representations. Without loss of generality, we assume a sigmoid function for the discriminative probability

$$q_t(y|\mathbf{x}; \mathsf{W}_t) = 1/(1 + \exp\{-y\mathbf{w}_t^{(1)} \cdot \phi(\mathbf{x}; \mathbf{w}_t^{(0)})\}),$$

where $\phi(\mathbf{x}; \mathbf{w}_t^{(0)})$ defines the feature extraction function for $\mathbf{x}$. Both $\mathbf{w}_t^{(1)}$ and $\mathbf{w}_t^{(0)}$ can be learned by the standard stochastic gradient descent algorithm via backpropagation to minimize a cross-entropy loss with an additional term on the pseudo-negatives:

$$\mathcal{L}(\mathsf{W}_t) = - \sum_{(\mathbf{x}_i, y_i) \in S}^{i=1..n} \ln q_t(y_i|\mathbf{x}_i; \mathsf{W}_t) - \sum_{(\mathbf{x}_i, -1) \in S_{pn}^t}^{i=n+1..n+tl} \ln q_t(-1|\mathbf{x}_i; \mathsf{W}_t). \tag{5}$$

---

**Algorithm 1** Outline of the reclassification-by-synthesis algorithm for discriminative classifier training.

---

**Input:** Given a set of training data $S = \{(\mathbf{x}_i, y_i), i = 1..n\}$ with $\mathbf{x} \in \mathbb{R}^m$ and $y \in \{-1, +1\}$.
**Initialization**: Obtain a reference distribution: $p_r^-(\mathbf{x}) = U(\mathbf{x})$ and train an initial CNN binary classifier $C^0$ on $S$, $q_0(y = +1|\mathbf{x})$. $S_{pn}^0 = \emptyset$. $U(\mathbf{x})$ is a zero mean Gaussian distribution.
**For** t=0..T
**1.** Update the model: $p_t^-(\mathbf{x}) = \frac{1}{Z_t} \frac{q_t(y=+1|\mathbf{x})}{q_t(y=-1|\mathbf{x})} p_r^-(\mathbf{x})$.
**2.** Synthesis-step: sample $l$ pseudo-negative samples $\mathbf{x}_i \sim p_t^-(\mathbf{x}), i = n+tl+1, ..., n+(t+1)l$ from the current model $p_t^-(\mathbf{x})$ using an SGD sampling procedure.
**3.** Augment the pseudo-negative set with $S_{pn}^{t+1} = S_{pn}^t \cup \{(\mathbf{x}_i, -1), i = n+tl+1, ..., n+(t+1)l\}$.
**4.** Reclassification-step: Update CNN classifier to $C^{t+1}$ on $S_e^{t+1} = S \cup S_{pn}^{t+1}$, resulting in $q_{t+1}(y = +1|\mathbf{x})$.
**5.** $t \leftarrow t + 1$ and go back to step 1 until convergence (e.g. no improvement on the validation set).
**End**

---

### 3.2.2 Synthesis-step

In the reclassification step, we obtain $q_t(y|\mathbf{x}; \mathsf{W}_t)$ which is then used to update $p_t^-(\mathbf{x})$ according to Eqn. (3):

$$p_t^-(\mathbf{x}) = \frac{1}{Z_t} \frac{q_t(y = +1|\mathbf{x}; \mathsf{W}_t)}{q_t(y = -1|\mathbf{x}; \mathsf{W}_t)} p_r^-(\mathbf{x}). \tag{6}$$

In the synthesis-step, our goal is to draw fair samples from $p_t^-(\mathbf{x})$ (fair samples refer to typical samples by a sampling process after convergence w.r.t the target distribution). In [40], various Markov chain Monte Carlo techniques [28] including Gibbs sampling and Iterated Conditional Modes (ICM) have been adopted, which are often slow. Motivated by the DeepDream code [32] and Neural Artistic Style work [9], we update a random sample $\mathbf{x}$ drawn from $p_r^-(\mathbf{x})$ by increasing $\frac{q_t(y=+1|\mathbf{x}; \mathsf{W}_t)}{q_t(y=-1|\mathbf{x}; \mathsf{W}_t)}$ using backpropagation. Note that the partition function (normalization) $Z_t$ is a constant that is not dependent on the sample $\mathbf{x}$. Let

$$g_t(\mathbf{x}) = \frac{q_t(y = +1|\mathbf{x}; \mathsf{W}_t)}{q_t(y = -1|\mathbf{x}; \mathsf{W}_t)} = \exp\{\mathbf{w}_t^{(1)} \cdot \phi(\mathbf{x}; \mathbf{w}_t^{(0)})\}, \tag{7}$$

and take its $\ln$, which is nicely turned into the logit of $q_t(y = +1|\mathbf{x}; \mathsf{W}_t)$

$$\ln g_t(\mathbf{x}) = \mathbf{w}_t^{(1)} \cdot \phi(\mathbf{x}; \mathbf{w}_t^{(0)}). \tag{8}$$

Starting from $\mathbf{x}$ drawn from $p_r^-(\mathbf{x})$, we directly increase $\mathbf{w}_t^{(1)T} \phi(\mathbf{x}; \mathbf{w}_t^{(0)})$ using stochastic gradient ascent on $\mathbf{x}$ via backpropagation, which allows us to obtain fair samples subject to Eqn. (6). Gaussian noise can be added to Eqn. (8) along the line of stochastic gradient Langevin dynamics [43] as

$$\Delta\mathbf{x} = \frac{\epsilon}{2} \nabla(\mathbf{w}_t^{(1)} \cdot \phi(\mathbf{x}; \mathbf{w}_t^{(0)})) + \eta$$

where $\eta \sim \mathcal{N}(0, \epsilon)$ is a Gaussian distribution and $\epsilon$ is the step size that is annealed in the sampling process.

**Sampling strategies**. When conducting experiments, we carry out several strategies using stochastic gradient descent algorithm (SGD) and SGD Lagenvin including: i) early-stopping for the sampling process after $\mathbf{x}$ becomes positive (aligned with contrastive divergence [4] where a short Markov chain is simulated); ii) stopping at a large confidence for $\mathbf{x}$ being positive, and iii) sampling for a fixed, large number of steps. Table 2 shows the results on these different options and no major differences in the classification performance are observed.

Building connections between SGD and MCMC is an active area in machine learning [43, 5, 30]. In [43], combining SGD and additional Gaussian noise under annealed stepsize results in a simulation of Langevin dynamics MCMC. A recent work [30] further shows the similarity between constant SGD and MCMC, along with analysis of SGD using momentum updates. Our progressively learned discriminative classifier can be viewed as carving out the feature space on $\phi(\mathbf{x})$, which essentially becomes an equivalent class for the positives; the volume of the equivalent class that satisfies the condition is exponentially large, as analyzed in [46]. The probability landscape of positives (equivalent class) makes our SGD sampling process not particularly biased towards a small limited modes. Results in Figure 2 illustrates that large variation of the sampled/synthesized examples.

## 3.3  Analysis

The convergence of $p_t^-(\mathbf{x}) \overset{t=\infty}{\Rightarrow} p^+(\mathbf{x})$ can be derived (see the supplementary material), inspired by the proof from [40]: $KL[p^+(\mathbf{x})||p_{t+1}^-(x)] \leq KL[p^+(\mathbf{x})||p_t^-(\mathbf{x})]$ where $KL$ denotes the Kullback-Leibler divergence and $p(\mathbf{x}|y=+1) \equiv p^+(\mathbf{x})$, under the assumption that classifier at $t+1$ improves over $t$.

**Remark**. Here we pay particular attention to the negative samples which live in a space that is often much larger than the positive sample space. For the negative training samples, we have $y_i = -1$ and $\mathbf{x}_i \sim Q^-(\mathbf{x})$, where $Q^-(\mathbf{x})$ is a distribution on the given negative examples in the original training set. Our reclassification-by-synthesis algorithm (Algorithm 1) essentially constructs a mixture model $\tilde{p}(\mathbf{x}) \equiv \frac{1}{T} \sum_{t=0}^{T-1} p_t^-(\mathbf{x})$ by sequentially generating pseudo-negative samples to augment our training set. Our new distribution for augmented negative sample set thus becomes $Q_{new}^-(\mathbf{x}) \equiv \frac{n}{n+Tl} Q^-(\mathbf{x}) + \frac{Tl}{n+Tl} \tilde{p}(\mathbf{x})$, where $\tilde{p}(\mathbf{x})$ encodes pseudo-negative samples that are confusing and similar to (but are not) the positives. In the end, adding pseudo-negatives might degrade the classification result since they become more and more similar to the positives. Cross-validation can be used to decide when adding more pseudo-negatives is not helping the classification task. How to better use the pseudo-negative samples that are increasingly faithful to the positives is an interesting topic worth further exploring. Our overall algorithm thus is capable of enhancing classification by self-generating confusing samples to improve CNN's robustness.

## 3.4  Multi-class classification

**One-vs-all**. In the above section, we discussed the binary classification case. When dealing with multi-class classification problems, such as MNIST and CIFAR-10, we will need to adapt our proposed reclassification-by-synthesis scheme to the multi-class case. This can be done directly using a one-vs-all strategy by training a binary classifier $C_i$ using the $i$-th class as the positive class and then combine the rest classes into the negative class, resulting in a total of $K$ binary classifiers. The training procedure then becomes identical to the binary classification case. If we have $K$ classes, then the algorithm will train $K$ individual binary classifiers with

$$< (\mathbf{w}_t^{(0)_1}, \mathbf{w}_t^{(1)_1}), ..., (\mathbf{w}_t^{(0)_K}, \mathbf{w}_t^{(1)_K}) > .$$

The prediction function is simply

$$f(\mathbf{x}) = \arg\max_k \exp\{\mathbf{w}_t^{(1)_k} \cdot \phi(\mathbf{x}; \mathbf{w}_t^{(0)_k})\}.$$

The advantage of using the one-vs-all strategy is that the algorithm can be made nearly identical to the binary case at the price of training $K$ different neural networks.

**Softmax function**. It is also desirable to build a single CNN classifier to perform multi-class classification directly. Here we propose a formulation to train an end-to-end multiclass classifier directly. Since we are directly dealing with $K$ classes, the pseudo-negative data set will be slightly different and we introduce negatives for each individual class by $S_{pn}^0 = \emptyset$ and:

$$S_{pn}^t = \{(\mathbf{x}_i, -k), k = 1, ..., K, i = n + (t-1) \times k \times l + 1, ..., n + t \times k \times l\}$$

Suppose we are given a training set $S = \{(\mathbf{x}_i, y_i), i = 1..n\}$ and $\mathbf{x} \in \mathbb{R}^m$ and $y \in \{1, .., K\}$. We want to train a single CNN classifier with

$$W_t = <\mathbf{w}_t^{(0)}, \mathbf{w}_t^{(1)_1}, ..., \mathbf{w}_t^{(1)_K}>$$

where $\mathbf{w}_t^{(0)}$ denotes the internal feature and parameters for the single CNN, and $\mathbf{w}_t^{(1)_k}$ denotes the top-layer weights for the $k$-th class. We therefore minimize an integrated objective function

$$\mathcal{L}(\mathbf{W}_t) = -(1-\alpha) \sum_{i=1}^n \ln \frac{\exp\{\mathbf{w}_t^{(1)_{y_i}} \cdot \phi(\mathbf{x}_i; \mathbf{w}_t^{(0)})\}}{\sum_{k=1}^K \exp\{\mathbf{w}_t^{(1)_k} \cdot \phi(\mathbf{x}_i; \mathbf{w}_t^{(0)})\}} + \alpha \sum_{i=n+1}^{n+t \times K \times l} \ln(1 + \exp\{\mathbf{w}_t^{(1)_{|y_i|}} \cdot \phi(\mathbf{x}_i; \mathbf{w}_t^0)\}) \quad (9)$$

The first term in Eqn. (9) encourages a softmax loss on the original training set $S$. The second term in Eqn. (9) encourages a good prediction on the individual pseudo-negative class generated for the $k$-th class (indexed by $|y_i|$ for $\mathbf{w}_t^{(1)|y_i|}$, e.g. for pseudo-negative samples belong to the $k$-th class, $|y_i| = |-k| = k$). $\alpha$ is a hyperparameter balancing the two terms. Note that we only need to build a single CNN sharing $\mathbf{w}_t^{(0)}$ for all the $K$ classes. In particular, we are not introducing additional model parameters here and we perform a direct $K$-class classification where the parameter setting is identical to a standard CNN multi-class classification task; to compare, an additional "not-real" class is created in [36] and the classification task there [36] thus becomes a $K + 1$ class classification.

# 4 Experiments

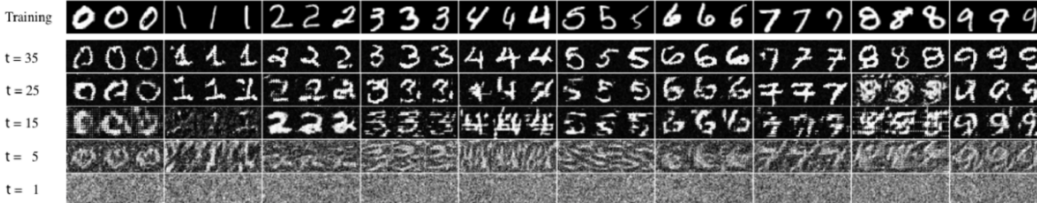

Figure 2: Synthesized pseudo-negatives for the MNIST dataset by our ICN classifier. The top row shows some training examples. As $t$ increases, our classifier gradually synthesize pseudo-negative samples that become increasingly faithful to the training samples.

We conduct experiments on three standard benchmark datasets, including MNIST, CIFAR-10 and SVHN. We use MNIST as a running example to illustrate our proposed framework using a shallow CNN; we then show competitive results using a state-of-the-art CNN classifier, ResNet [12] on MNIST, CIFAR-10 and SVHN. In our experiments, for the reclassification step, we use the SGD optimizer with mini-batch size of 64 (MNIST) or 128 (CIFAR-10 and SVHN) and momentum equal to 0.9; for the synthesis step, we use the Adam optimizer [17] with momentum term $\beta_1$ equal to 0.5. All results are obtained by averaging multiple rounds.

**Training and test time**. In general, the training time for ICN is around double that of the baseline CNNs in our experiments: 1.8 times for MNIST dataset, 2.1 times for CIFAR-10 dataset and 1.7 times for SVHN dataset. The added overhead in training is mostly determined by the number of generated pseudo-negative samples. For the test time, ICN introduces no additional overhead to the baseline CNNs.

## 4.1 MNIST

We use the standard MNIST [24] dataset, which consists of $55,000$ training, $5,000$ validation and $10,000$ test samples. We adopt a simple network, containing 4 convolutional layers, each having a $5 \times 5$ filter size with 64, 128, 256 and 512 channels, respectively. These convolutional layers have stride 2, and no pooling layers are used. LeakyReLU activations [29] are used after each convolutional layer. The last convolutional layer is flattened and fed into a sigmoid output (in the one-vs-all case).

In the reclassification step, we run SGD (for 5 epochs) on the current training data $S_e^t$, including previously generated pseudo-negatives. Our initial learning rate is 0.025 and is decreased by a factor of 10 at $t = 25$. In the synthesis step, we use the backpropagation sampling process as discussed in Section 3.2.2. In Table 2, we compare different sampling strategies. Each time we synthesize a fixed number (200 in our experiments) of pseudo-negative samples.

Table 1: Test errors on the MNIST dataset. We compare our ICN method with the baseline CNN, Deep Belief Network (DBN) [14], and CNN w/ Label Smoothing (LS) [39]. Moreover, the two-step experiments combining CNN + GDL [40] and combining CNN + DCGAN [35] are also reported, and see descriptions in text for more details.

| Method | One-vs-all (%) | Softmax (%) |
|---|---|---|
| DBN | - | 1.11 |
| CNN (baseline) | 0.87 | 0.77 |
| CNN w/ LS | - | 0.69 |
| CNN + GDL | 0.85 | - |
| CNN + DCGAN | 0.84 | - |
| ICN-noise (ours) | 0.89 | 0.77 |
| ICN (ours) | 0.78 | 0.72 |

We show some synthesized pseudo-negatives from the MNIST dataset in Figure 2. The samples in the top row are from the original training dataset. ICN gradually synthesizes pseudo-negatives, which are increasingly faithful to the original data. Pseudo-negative samples will be continuously used while improving the classification result.

**Comparison of different sampling strategies.** We perform SGD and SGD Langevin (with injected Gaussians), and try several options via backpropagation for the sampling strategies. Option 1: early-stopping once the generated samples are classified as positive; option 2: stopping at a high confidence for samples being positive; option 3: stopping after a large number of steps. Table 2 shows the results and we do not observe significant differences in these choices.

Table 2: Comparison of different sampling strategies in the synthesis step in ICN.

| Sampling Strategy | One-vs-all (%) | Softmax (%) |
|---|---|---|
| SGD (option 1) | 0.81 | 0.72 |
| SGD Langevin (option 1) | 0.80 | 0.72 |
| SGD (option 2) | 0.78 | 0.72 |
| SGD Langevin (option 2) | 0.78 | 0.74 |
| SGD (option 3) | 0.81 | 0.75 |
| SGD Langevin (option 3) | 0.80 | 0.73 |

**Ablation study**. We experiment using random noise as synthesized pseudo-negatives in an ablation study. From Table 1, we observe that our ICN outperforms the CNN baseline and the ICN-noise method in both one-vs-all and softmax cases.

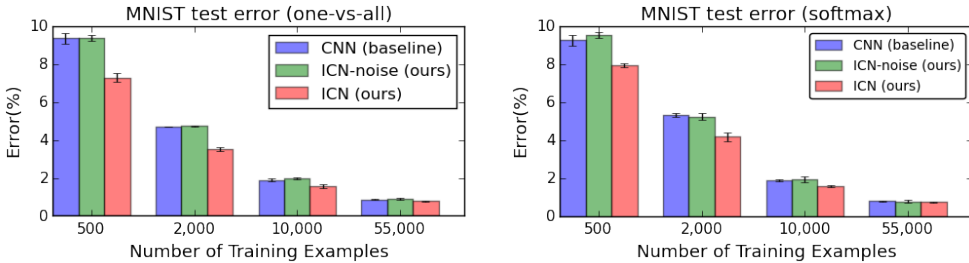

Figure 3: MNIST test error against the number of training examples (std dev. of the test error is also displayed). The effect of ICN is more clear when having fewer training examples.

**Effects on varying training sizes.** To better understand the effectiveness of our ICN method, we carry out an experiment by varying the number of training examples. We use training sets with different sizes including 500, 2000, 10000, and 55000 examples. The results are reported in Figure 3. ICN is shown to be particularly effective when the training set is relatively small, since ICN has the capability to synthesize pseudo-negatives by itself to aid training.

**Comparison with GDL and GAN.** GDL [40] focuses on unsupervised learning; GAN [10] and DCGAN [35] show results for unsupervised learning and semi-supervised classification. To apply GDL and GAN to the supervised classification setting, we design an experiment to perform a two-step implementation. For GDL, we ran the GDL code [40] and obtained the pseudo-negative samples for each individual digit; the pseudo-negatives are then used as augmented negative samples to train individual one-vs-all CNN classifiers (using an identical CNN architecture to ICN for a fair comparison), which are combined to form a multi-class classifier in the end. To compare with DCGAN [35], we follow the same procedure: each generator trained by DCGAN [35] using the TensorFlow implementation [16] was used to generate positive samples, which are then augmented to the negative set to train the individual one-vs-all CNN classifiers (also using an identical CNN architecture to ICN), which are combined to create the overall multi-class classifier. CNN+GDL achieves a test error of 0.85% and CNN+DCGAN achieves a test error of 0.84% on the MNIST dataset, whereas ICN reports an error of 0.78% using the same CNN architecture. As the supervised learning task was not directly specified in DCGAN [35], some care is needed to design the optimal setting to utilize the generated samples from DCGAN in the two-step approach (we made attempts to optimize the results). GDL [40] can be made into a discriminative classifier by utilizing the given negative samples first but boosting [7] with manually designed features was adopted which may not produce competitive results as CNN classifier does. Nevertheless, the advantage of ICN being an integrated end-to-end supervised learning single-model framework can be observed.

To compare with generative model based deep learning approach, we report the classification result of DBN [14] in Table 1. DBN achieves a test error of 1.11% using the softmax function. We also compare with Label Smoothing (LS), which has been used in [39] as a regularization technique by encouraging the model to be less confident. In LS, for a training example with ground-truth label, the label distribution is replaced with a mixture of the original ground-truth distribution and a fixed distribution. LS achieves a test error of 0.69% in the softmax case.

In addition, we also adopt ResNet-32 [13] (using the softmax function) as another baseline CNN model, which achieves a test error of $0.50\%$ on the MNIST dataset. Our ResNet-32 based ICN achieves an improved result of $0.47\%$.

**Robustness to external adversarial examples.** To show the improved robustness of ICN in dealing with confusing and challenging examples, we compare the baseline CNN with our ICN classifier on adversarial examples generated using the "fast gradient sign" method from [11]. This "fast gradient sign" method (with $\epsilon = 0.25$) can cause a maxout network to misclassify $89.4\%$ of adversarial examples generated from the MNIST test set [11]. In our experiment, we set $\epsilon = 0.125$. Starting with $10,000$ MNIST test examples, we first determine those which are correctly classified by the baseline CNN in order to generate adversarial examples from them. We find that $5,111$ generated adversarial examples successfully fool the baseline CNN, however, only $3,134$ of these examples can fool our ICN classifier, which is a $38.7\%$ reduction in error against adversarial examples. Note that the improvement is achieved without using any additional training data, nor knowing a prior about how these adversarial examples are generated by the specific "fast gradient sign method" [11]. On the contrary, of the $2,679$ adversarial examples generated from the ICN classifier side that fool ICN using the same method, $2,079$ of them can still fool the baseline CNN classifier. This two-way experiment shows the improved robustness of ICN over the baseline CNN.

## 4.2 CIFAR-10

The CIFAR-10 dataset [18] consists of $60,000$ color images of size $32 \times 32$. This set of $60,000$ images is split into two sets, $50,000$ images for training and $10,000$ images for testing. We adopt ResNet [13] as our baseline model [45]. For data augmentation, we follow the standard procedure in [26, 25, 13] by augmenting the dataset by zero-padding 4 pixels on each side; we also perform cropping and random flipping. The results are reported in Table 3. In both one-vs-all and softmax cases, ICN outperforms the baseline ResNet classifiers. Our proposed ICN method is orthogonal to many existing approaches which use various improvements to the network structures in order to enhance the CNN performance. We also compare ICN with Convolutional

Table 3: Test errors on the CIFAR-10 dataset. In both one-vs-all and softmax cases, ICN shows improvement over the baseline ResNet model. The result of convolutional DBN is from [19].

| Method | One-vs-all (%) | Softmax (%) |
|---|---|---|
| w/o Data Augmentation | | |
| Convolutional DBN | - | 21.1 |
| ResNet-32 (baseline) | 13.44 | 12.38 |
| ResNet-32 w/ LS | - | 12.65 |
| ResNet-32 + DCGAN | 12.99 | - |
| ICN-noise (ours) | 13.28 | 11.94 |
| ICN (ours) | 12.94 | 11.46 |
| w/ Data Augmentation | | |
| ResNet-32 (baseline) | 6.70 | 7.06 |
| ResNet-32 w/ LS | - | 6.89 |
| ResNet-32 + DCGAN | 6.75 | - |
| ICN-noise (ours) | 6.58 | 6.90 |
| ICN (ours) | 6.52 | 6.70 |

DBN [19], ResNet-32 w/ Label Smoothing (LS) [39] and ResNet-32+DCGAN [35] methods as described in the MNIST experiments. LS is shown to improve the baseline but is worse than our ICN method in most cases except for the MNIST dataset.

## 4.3 SVHN

We use the standard SVHN [33] dataset. We combine the training data with the extra data to form our training set and use the test data as the test set. No data augmentation has been applied. The result is reported in Table 4. ICN is shown to achieve competitive results.

Table 4: Test errors on the SVHN dataset.

| Method | Softmax (%) |
|---|---|
| ResNet-32 (baseline) | 2.01 |
| ResNet-32 w/ LS | 1.96 |
| ResNet-32 + DCGAN | 1.98 |
| ICN-noise (ours) | 1.99 |
| ICN (ours) | 1.95 |

## 5 Conclusion

In this paper, we have proposed an introspective convolutional nets (ICN) algorithm that performs internal introspection. We observe performance gains within supervised learning using state-of-the-art CNN architectures on standard machine learning benchmarks.

**Acknowledgement** This work is supported by NSF IIS-1618477, NSF IIS-1717431, and a Northrop Grumman Contextual Robotics grant. We thank Saining Xie, Weijian Xu, Fan Fan, Kwonjoon Lee, Shuai Tang, and Sanjoy Dasgupta for helpful discussions.

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
