[Supplementary Material]

# Introspective Classification with Convolutional Nets Supplementary Material

**Long Jin**
UC San Diego
longjin@ucsd.edu

**Justin Lazarow**
UC San Diego
jlazarow@ucsd.edu

**Zhuowen Tu**
UC San Diego
ztu@ucsd.edu

## A  Proof of the convergence of $p_t^-(\mathbf{x}) \overset{t=\infty}{\rightarrow} p^+(\mathbf{x})$

The convergence of $p_t^-(\mathbf{x}) \overset{t=\infty}{\rightarrow} p^+(\mathbf{x})$ can be derived, inspired by the proof from [1]: $KL[p^+(\mathbf{x})||p_{t+1}^-(x)] \leq KL[p^+(\mathbf{x})||p_t^-(\mathbf{x})]$ where $KL$ denotes the Kullback-Leibler divergence and $p(x|y=+1) \equiv p^+(x)$, under the assumption that classifier at $t+1$ improves over $t$.

Proof:

$$p_t^-(\mathbf{x}) = \frac{1}{Z_t} \frac{q_t(y=+1|\mathbf{x})}{q_t(y=-1|\mathbf{x})} p_r^-(\mathbf{x}),$$

and

$$p_{t+1}^-(\mathbf{x}) = \frac{1}{Z_{t+1}} \frac{q_{t+1}(y=+1|\mathbf{x})}{q_{t+1}(y=-1|\mathbf{x})} p_r^-(\mathbf{x}).$$

$$p_{t+1}^-(\mathbf{x}) = \frac{1}{H_{t+1}} \frac{\exp\{\mathbf{w}_{t+1}^{(1)T}\phi(\mathbf{x};\mathbf{w}_{t+1}^{(0)})\}}{\exp\{\mathbf{w}_t^{(1)T}\phi(\mathbf{x};\mathbf{w}_t^{(0)})\}} p_t^-(\mathbf{x})$$

where

$$H_{t+1} = \int \frac{\exp\{\mathbf{w}_{t+1}^{(1)T}\phi(\mathbf{x};\mathbf{w}_{t+1}^{(0)})\}}{\exp\{\mathbf{w}_t^{(1)T}\phi(\mathbf{x};\mathbf{w}_t^{(0)})\}} p_t^-(\mathbf{x})d\mathbf{x}$$

$$
\begin{aligned}
&KL[p^+(\mathbf{x})||p_t^-(\mathbf{x})] - KL[p^+(\mathbf{x})||p_{t+1}^-(\mathbf{x})]\\
=\ & \int p^+(\mathbf{x}) \ln \left( \frac{1}{H_{t+1}} \frac{\exp\{\mathbf{w}_{t+1}^{(1)T}\phi(\mathbf{x};\mathbf{w}_{t+1}^{(0)})\}}{\exp\{\mathbf{w}_t^{(1)T}\phi(\mathbf{x};\mathbf{w}_t^{(0)})\}} p_t^-(\mathbf{x}) \right) d\mathbf{x} - \int p^+(\mathbf{x}) \ln[p_t^-(\mathbf{x})]dx\\
=\ & \int p^+(\mathbf{x}) \ln \frac{1}{H_{t+1}} d\mathbf{x} + \int p^+(\mathbf{x}) \ln \frac{\exp\{\mathbf{w}_{t+1}^{(1)T}\phi(\mathbf{x};\mathbf{w}_{t+1}^{(0)})\}}{\exp\{\mathbf{w}_t^{(1)T}\phi(\mathbf{x};\mathbf{w}_t^{(0)})\}} d\mathbf{x}\\
=\ & \ln \frac{1}{H_{t+1}} + \int p^+(\mathbf{x}) \ln \frac{\exp\{\mathbf{w}_{t+1}^{(1)T}\phi(\mathbf{x};\mathbf{w}_{t+1}^{(0)})\}}{\exp\{\mathbf{w}_t^{(1)T}\phi(\mathbf{x};\mathbf{w}_t^{(0)})\}} d\mathbf{x} \geq 0.
\end{aligned}
$$

Since $H_{t+1} = \int \frac{\exp\{\mathbf{w}_{t+1}^{(1)T}\phi(\mathbf{x};\mathbf{w}_t^{(0)})\}}{\exp\{\mathbf{w}_t^{(1)T}\phi(\mathbf{x};\mathbf{w}_t^{(0)})\}} p_t^-(\mathbf{x})d\mathbf{x} \leq 1$ and $\int p^+(\mathbf{x}) \ln \frac{\exp\{\mathbf{w}_{t+1}^{(1)T}\phi(\mathbf{x};\mathbf{w}_t^{(0)})\}}{\exp\{\mathbf{w}_t^{(1)T}\phi(\mathbf{x};\mathbf{w}_t^{(0)})\}} d\mathbf{x} \geq 0$. Given the training data and the previously generated pseudo-negative samples are all retained in each step, we assume that the classifier at $t+1$ improves over that at $t$. This shows that $p_{t+1}^-(\mathbf{x})$ converges to $p(\mathbf{x}|y=+1)$ and the convergence rate depends on the classification error at each step.

## References

[1] Z. Tu. Learning generative models via discriminative approaches. In *CVPR*, 2007.