[Reviews · NeurIPS 2017]

Reviewer 1



The paper proposes a technique to improve the test accuracy of a discriminative model, by synthesizing additional negative input examples during the training process of the model. The negative example generation process has a Bayesian motivation, and is realized by "optimizing" for images (starting from random Gaussian noise) to maximize the probability of a given class label, a la DeepDream or Neural Artistic Style. These generated examples are added to the training set, and training is halted based on performance on a validation set. Experiments demonstrate that this procedure yields (very modest) improvements in test accuracy, and additionally provides some robustness against adversarial examples. The core idea is quite elegant, with an intuitive picture of using the "hard" negatives generated by the network to tighten the decision boundaries around the positive examples. The authors provide some theoretical justification showing that the distribution of these negatives examples converges to the distribution of positive examples. While I find the approach intriguing, there are a few shortcomings. The improvement in classification accuracy demonstrated in the experiments is relatively small for the datasets shown, although the MNIST demonstration with varying sample sizes is very encouraging. A more complex dataset (e.g. ImageNet) would be very interesting to see, but potentially quite expensive to train... I'm unable to determine how many pseudo-negative samples are generated in each round (it says "e.g. 200" at one point; is it 200?), which affects the practicality of the approach (how much slower is training?). While the Bayesian motivation is nice, it's not clear to me exactly what it means to sample from the distribution of negative examples using SGD. I'm not actively familiar with the references listed tying SGD to MCMC; if there is in fact a meaningful interpretation of the proposed sampling procedure that justifies the vague (but repeated) use of the term "fair sample", it would be nice to include a more explicit summary in the text. Finally, it would have been nice to see comparisons against DCGAN for CIFAR10 and SVHN as well, as well as a simple baseline of label smoothing (instead of using one-hot target labels, use a mixture of e.g. .9 * one-hot + .1 * uniform). Label smoothing has been reported (e.g., in https://arxiv.org/pdf/1512.00567.pdf) to yield comparably sized improvements, essentially by encouraging the model to be less confident. One explanation for the success of the ICN approach could simply be that by training on negative samples that are close to the positive samples, the model is forced to be less confident overall, reducing overfitting. Comparing against label smoothing could help tease out whether that explanation is sufficient (and if so, label smoothing is substantially simpler and cheaper, if a bit of a hack). Despite the above criticisms, I think the paper is well written and could be a valuable contribution; I recommend acceptance.

Reviewer 2



Authors present the so called introspective convolutional network (ICN), a deep learning model that is able, using a single model, to generate negative example in order to improve the network training and in turn its classification performances. The proposed model is compared with a state-of-the-art discriminative convolutional neural network (CNN) for classification, namely ResNet, and with other two networks sharing a similar approach with the ICN. Classification scores over well known benchmark dataset confirms a performance gain. The paper is well written and the proposed method is interesting in my opinion. I have one question: did you consider to compare your ICN against a deep belief network (DBN)? a DBN is in fact a probabilistic generative model also used for classification that, during the pre-trainining phase uses a generative model of the input data by means of a stack of Restricted Boltzmann machine and then, in the fine tuning, it is trained such a multilayer perceptron using gradient descent and backpropagation. I think that a comparison with that network could further improve the quality of the paper. Minor remarks: Which programming languages or libraries did you use for your network? Please clarify and let the source code be avilable, for example on github What are the computation time? please add a table Figure 1 is too small, please increase its size

Reviewer 3



The key idea of ICN is to: 1) train a base classifier and use it to generate pseudo-negative samples; 2) add the generated samples to the original training dataset and fit a new classifier on it. The main drawback of this approach is that it is very easy for the classifier to overfit on the positive samples, as indicated in Figure 1. Although the authors give convergence analysis (line 212), the assumption (that the classifier trained on new samples always improves over the old classifier) is too strong. As a result, the reviewer is not convinced by the proposed algorithm. The experimental results are not convincing as well. The baseline results reported in the experiments are far from the state of the art (SOTA) results, e.g., 0.77 vs 0.21 (SOTA) on the MNIST dataset. The proposed algorithm is a meta-algorithm, which does not depend on the underlying architecture of the base classifier. Therefore, it is more convincing if the authors use the SOTA (or close to SOTA) neural classifier architecture as the baseline, and then see if using the idea of ICN could improve on it. A key assumption in the formulation (line 106) is that the priors are equal: p(y=1)=p(y=-1). This can be problematic; as most real datasets have unbalanced class labels. The multi-class formulation (line 244) pushes the classifier away from the generated pseudo-negative samples. However, the pseudo-negative samples are generated for each class. A natural question is what if the pseudo-negative sample of class-1 is the pseudo-positive sample of class-2? In that case, pushing the classifier away from this sample could degrade the classifier accuracy for class-2.